# Find the Needle in the Haystack, Then Find It Again: Replication and Validation in the ‘Omics Era

**DOI:** 10.3390/metabo10070286

**Published:** 2020-07-12

**Authors:** Wei Perng, Stella Aslibekyan

**Affiliations:** 1Department of Epidemiology, Colorado School of Public Health, University of Colorado Denver Anschutz Medical Campus, Aurora, CO 80045, USA; 2Lifecourse Epidemiology of Adiposity and Diabetes (LEAD) Center, University of Colorado Denver Anschutz Medical Campus, Aurora, CO 80045, USA; 3Department of Epidemiology, University of Alabama at Birmingham, Birmingham, AL 35294, USA; saslibek@uab.edu

**Keywords:** replication, validation, ‘omics, genomics, metabolomics, integrative ‘omics

## Abstract

Advancements in high-throughput technologies have made it feasible to study thousands of biological pathways simultaneously for a holistic assessment of health and disease risk via ‘omics platforms. A major challenge in ‘omics research revolves around the reproducibility of findings—a feat that hinges upon balancing false-positive associations with generalizability. Given the foundational role of reproducibility in scientific inference, replication and validation of ‘omics findings are cornerstones of this effort. In this narrative review, we define key terms relevant to replication and validation, present issues surrounding each concept with historical and contemporary examples from genomics (the most well-established and upstream ‘omics), discuss special issues and unique considerations for replication and validation in metabolomics (an emerging field and most downstream ‘omics for which best practices remain yet to be established), and make suggestions for future research leveraging multiple ‘omics datasets.

“The natural scientist is concerned with a particular kind of phenomenon… he has to confine himself to that which is reproducible… I do not claim that the reproducible by itself is more important than the unique. But I do claim that the unique exceeds the treatment by scientific method.” Wolfgang Pauli

Advances in high-throughput technologies have ushered in an era of “big data”, wherein researchers are no longer relegated to investigating single biological pathways. Rather, scientists grapple with a newfound ability to study thousands of pathways simultaneously for a more holistic understanding of (patho)physiology. The analysis and interpretation of large, high-dimensional biological data is spearheaded by the ‘omics fields comprising genomics, epigenomics, transcriptomics, proteomics, and metabolomics. The oldest and best known ‘omics is genomics, or the comprehensive study of DNA within an organism’s genome. Genomics is complemented by epigenomics, the study of the complete set of potentially reversible epigenetic modifications made to the DNA sequence. The next ‘omics layer is transcriptomics, the systematic study of RNA transcripts and gene expression. Following transcriptomics is proteomics, the study of proteins and their role in carrying out basic life processes. At the end of the ‘omics cascade is metabolomics, the study of low-molecular-weight compounds in biological tissues and fluids.

While much of the data generated by the ‘omics fields in the last two decades have been qualitative or semi-quantitative in nature, rigorous quantitative analysis is key to formulating and testing novel hypotheses with the greatest potential to yield actionable next steps in the health sciences. Yet, a major challenge of quantitative ‘omics research revolves around the reproducibility of findings—a feat that lies in balancing false-positive findings (i.e., type 1 error) with generalizability (i.e., results that are too specific to a given study sample due to systematic bias [1] and/or model overfitting).

In this narrative review, we discuss the concepts of validation and replication in the context of genomics and metabolomics. Genomics is highly relevant to reproducibility given that many accepted approaches to ‘omics analyses were established in the field of genomics. At the other end of the spectrum, metabolomics is the newest ‘omics field, and the most downstream component of the ‘omics cascade. As such, the metabolome is closest to phenotype (and thus, perhaps most useful as a biomarker of disease risk or prognosis) but also exhibits the most inter- and intra-individual variability, thereby making replication and validation more challenging. We begin by defining key terminology, followed by a presentation and discussion of commonly used approaches, summarize guidelines (Table 1) for rigorous follow-up analyses of initial discoveries, and make recommendations for ways in which the methodology discussed herein may be applied to metabolomics research.

## 1. Terminology

While the practice of replication and validation is considered to be de rigueur in genetic epidemiology and genomics, there remains confusion regarding the two commonly used terms. Confirmation studies often seek to “replicate” or “validate” results without defining what this attempt comprises. As pointed out by Igl et al. [2], investigators have used the two terms interchangeably to describe when the same disease-biomarker associations are detected in two independent study populations [3], while other studies exclusively use the term “validation” [4] or “replication” [5]. Cavalier use of terminology is further compounded by varying degrees of similarity between the original sample (i.e., the discovery sample) vs. the replication/validation sample. For example, epidemiologists may conduct validation/replication studies among study populations of different racial backgrounds to interrogate the utility of a biomarker across the full spectrum of human diversity. Alternatively, laboratory scientists may be interested in using different technologies to measure the same biological relationships in the same sample to corroborate legitimacy of a new technique vs. a gold standard. These inconsistencies make it such that discrepancies in results between the discovery and replication/validation samples may not only be explained by random variation, but also by systematic differences. Such issues complicate the interpretation of results – especially when confirmation of the initial findings fails. Thus, in this section, we define replication and validation and provide examples of each in the field of metabolomics.

### 1.1. Replication

Replication is the more straightforward of the two types of confirmatory studies. When replication is the goal, the original and confirmatory study samples should be independent but similar (i.e., similar sex, age, and race/ethnic distribution among the two samples) to reduce systematic differences, and the laboratory method used should be identical. Under these conditions, discrepancies in results are presumably due to random variation. Clarke et al. [6] describe the gold standard for replication as a test of the association of interest under identical circumstances using an independent sample. In the case of ‘omics analyses that include both untargeted (hypothesis-generating) and targeted (hypothesis-driven) approaches, replication entails using the same biological and statistical approaches in the discovery and replication samples (e.g., the choice of analytical platform, timing of measurements, quality control procedures, statistical models, and other design variables), and success is defined by the reproducibility of findings across the two populations. In practice, however, achieving such parity of conditions presents a nontrivial challenge, especially when multiple cohorts, investigators, and measurement periods are involved.

Replication is particularly challenging for metabolomics, given that circulating metabolites are subject to metabolic flux (up-regulation or down-regulation of biochemical pathways), even in the absence of alterations to upstream ‘omics that influence metabolism. This phenomenon contributes not only to inter-individual variability (which is relatively easy to control or account for in the study design or statistical analysis), but also intra-individual variability in compounds due to factors for which researchers do not typically collect information, such as time of day of data collection [7], season [8], and circadian or hormonal cycles [9]. Therefore, most efforts to confirm ‘omics findings fall under the umbrella of validation (discussed in the next section) rather than true replication. We note that in the absence of true replication, there are specific examples in the literature where certain metabolite profiles have emerged as being relevant to specific biological conditions across different study populations and age groups, and using different laboratory platforms and statistical techniques. One such example is that of metabolites involved in branched chain amino acid and acylcarnitine metabolic pathways which have been identified as correlates [10,11,12,13,14] and prospective risk factors for cardiometabolic biomarkers [15], as well as biomarkers that are evident up to a decade prior to development of obesity-related chronic disease like type 2 diabetes [16,17]. While these studies were conducted without the a priori intent of replication, consistency in findings within the broader literature provide confidence that such findings are not spurious.

### 1.2. Validation

The term “validation” is less straightforward, and refers to a few distinct concepts. The most established meaning of the term predates the era of ‘omics discovery, and refers to comparisons of specific laboratory techniques against a gold standard and is usually conducted on the same population. A hypothesis-free association study may yield several associations of interest (“hits”) in the discovery sample. The practice to date has been to validate such hits using the more expensive and time-consuming methods. In genomics, this approach has historically involved Sanger sequencing to confirm the nucleotide sequence. In metabolomics, this process involves implementation of targeted metabolomics assays to measure absolute concentrations of metabolites on a specific biochemical pathway following an initial associational discovery analysis of data generated on an untargeted metabolomics platform. In such a scenario, the discovery and validation samples are from the same population so that the only variation is introduced by the laboratory technique rather than systematic differences between populations. This form of validation is best suited for novel laboratory techniques or platforms and is less applicable to large-scale population-based studies.

The second distinction is internal vs. external validation. Internal validation is used in the development of predictive models that seek to identify features most strongly associated with the disease or biological condition of interest. Building the predictive model requires a training dataset to identify a set or a group of features that account for maximal variability in the biological condition. After training, the model is tested on an independent dataset or multiple resampled datasets to assess model validity. Techniques like bootstrapping and cross-validation optimize the model and check for overfitting. Examples of such approaches include random forests [18,19], a classification method that operates by constructing a multitude of decision trees with the goal of reducing variance; partial least squares-discriminant analysis [20], which relates the independent variable (e.g., metabolite concentration) to the response vector (the biological condition of interest) with a regression model; and least absolute shrinkage and selection operator (LASSO) [21]. A critical limitation of predictive modeling is the tendency to overfit data; thus, internal validation in an independent sample is of paramount importance. On the other hand, external validation refers to a study where the confirmation sample is a different population from that of the discovery sample. Differences between the two samples may be with respect to ethnic background or other sociodemographic characteristics, recruitment or sampling strategy, ascertainment of key variables, or time-point of investigation. In such scenarios, differences in findings may be explained by both random and systematic variability. Thus, large-scale, agnostic hypothesis-generating studies that achieve reproducibility using this approach indicate a robust association of the ‘omics feature with the biological condition and supports an argument for causality. In contrast, failure to reproduce findings from the discovery sample in a sufficiently different population does not necessarily imply false positivity of the initial finding, but, rather, a need to attempt external validation again. This strategy is currently the most prevalent in the field due to the diversity of existing cohorts; therefore, we will focus our discussion on this approach to validation.

## 2. A Deeper Dive into Validation

### 2.1. Selecting Features for Validation

#### 2.1.1. Correcting for Multiple Comparisons

With the advent of genome-wide association studies (GWAS) and GWAS consortia like the U.K. Biobank [22] or industry cohorts (e.g., 23andMe), the sample size within which scientists may explore single nucleotide polymorphism (SNP)–trait associations increased upwards of a thousand-fold, thereby providing greater statistical power to detect true associations. Additionally, these consortia served as large, diverse populations for validation of cohort-specific findings. As a consequence, the trend of irreproducible SNP–trait associations that plagued early genomics studies has been sharply reversed [23]. In metabolomics, researchers have taken approaches that mirror those of GWAS by examining associations of individual metabolites with a predictor or outcome and correcting for multiple comparisons in metabolome-wide association studies (MWAS). Yet, because metabolomics data exhibit high collinearity among compounds on similar biochemical pathways, there is greater likelihood for biological and statistical correlation among metabolites than SNPs, and conventional approaches to multiple comparisons correction are likely too stringent. Arguably, the most popular approach is to control the family wise error rate, which is the probability of one or more significant results under the null hypothesis of no association. For a high-dimensional dataset, the simplest way to do this is to control for the number of molecular variables using the Bonferroni correction. However, these methods assume independence among the molecular variables of interest. Less conservative approaches include the Benjamini–Hochberg false discovery rate (FDR) procedure [24], which sets a threshold for statistical significance based on the distribution of *p*-values, given an allowable error rate. The FDR procedure has been successfully applied to datasets with a relatively large number of true positive associations, such as gene expression data. However, the use of FDR procedures for MWAS is compounded by the correlated nature of individual metabolic pathways (and super- and sub-pathways, for that matter), and lack of available annotations for much of the human metabolome. Thus, assignment of molecular formulas to spectral peaks during the data processing step of metabolomics pipelines does not yield comprehensive structural identifications that are critical to biological understanding [25]. Suffice it to say, identification of suitable multiple testing thresholds in MWAS is a Herculean task and an area of active investigation that involves crosstalk among laboratory scientists, bioinformaticians, and statisticians. These efforts, in conjunction with metabolomics consortia—so far, the primary one being the COnsortium of METabolomics Studies (COMETS)—will provide valuable resources that may be used to validate initial cohort-specific findings and to test/compare the utility of appropriate multiple comparisons correction procedures.

Around the time when genetic consortia rose to prominence, it also became commonplace to reduce the dimensionality of big data by aggregating the molecular variables into pathways to first identify pathways of interest prior to assessment of individual features within the pathways. Dimension reduction is especially attractive and biologically intuitive for metabolomics analyses. Widely-used multivariate procedures include unsupervised principal components analysis [10,13,26] and weighted correlation network analysis [27] to consolidate metabolites into clusters prior to regression analysis. However, it must be cautioned that such procedures can lead to higher rates of false-positive findings [28], since investigators typically implement multiple testing corrections at the step of identifying metabolite clusters of interest, but not when examining associations with respect to individual metabolites. Additionally, because dimension reduction procedures often require complete data, the investigator must accept heroic assumptions about missing values. The assumptions are made following decisions regarding whether missing values are biologically meaningful (e.g., indicative of lower exposure to an environmental factor and thus, lower exposure of a metabolite affected by that factor) or whether missingness is a result of errors in laboratory procedures. If investigators suspect the former to be the case, then imputation procedures are implemented prior to use of dimension reduction algorithms. Common algorithms include setting missing values to ½ the lowest detected value when the percent of missing values for a compound is below an arbitrary threshold (typically < 20%); or more sophisticated approaches that yield a range of values, like K-nearest neighbor algorithm, when the proportion of missingness is higher [29]. Alternatively, if metabolite missingness is suspected to be due to issues with the laboratory procedure, then the investigator may opt to omit that particular compound from the analysis altogether. The advantages, disadvantages, and appropriateness of these approaches are discussed in detail elsewhere [30]. The key point here is that given the interconnectedness of metabolic pathways, such decisions are not trivial as their consequences have reverberations for subsequent multivariate analyses and interpretation of results.

#### 2.1.2. Metrics for Selection of Findings to Follow-Up Studies

The traditional approach to validating ‘omics findings begins with identifying a set of features that are associated with biological trait of interest at a predefined threshold of statistical significance, then further exploring the feature of interest in follow-up analyses. As discussed in the previous section, conservative corrections for multiple comparisons improve the reproducibility of ‘omics findings, but because metabolomics features are not independent, such correction is overly conservative. Moreover, corrections based on the number of aggregate pathways rather than single features have been shown to be insufficient [28], leaving room for development of other approaches. Thus, an ideal approach—particularly for validating discovery analyses—is to first implement a dimension reduction procedure to identify aggregate pathways of interest while controlling type 1 error using an appropriate method for the multivariate procedure implemented (e.g., Bonferroni correction is acceptable for procedures that derive uncorrelated latent variables like principal components analysis, whereas the more lenient FDR correction may be more appropriate for procedures like WGCNA where metabolite networks may be correlated), then explore associations with respect to individual metabolites. On balance, acceptable false discovery rates depend on the research question and the relative potential harm caused by false-positive (irreproducible) vs. false-negative (missed) findings. Such considerations will require investigators to weigh the costs vs. benefits of false-negative vs. false-positive findings, which will depend on the biological outcome of interest.

Of note, the most widely-used approach to selecting signals for further follow up (whether replication or validation) relies solely on *p*-values, leaving the analysis vulnerable to false-positive findings even after adjustment for multiple comparisons [31]. One remedy is to consider not only the *p*-value, but also the magnitude of association (i.e., effect size). This notion is fundamental to Sir Bradford Hill’s first guideline for evaluating causality: the strength of an association provides support for a causal effect of an exposure on disease risk [32]. This is due, in part, to the fact that large effect sizes require a large amount of bias to attenuate to the null and therefore are robust to both random and systematic error (the interested reader may peruse the article by Fedek et al. [33] for additional insights into application of Hill’s causal guidelines in the age of high-dimensional data).

In practice, there are a number of statistical techniques that focus on identifying a set of predictors from a high-dimensional dataset on the basis of magnitude of association, in addition to statistical significance. For example, the simulation extrapolation (SIMEX) algorithm has been successfully used for metabolomics data [34] estimates regression coefficients for the strongest set of predictors via a simulation step that generates multiple datasets with varying degrees of random error in the original data that plague laboratory procedures (e.g., residual batch effects; uncontrolled and/or unmeasured variation in technical methods; noise), followed by an extrapolation step that corrects for the additive measurement error via a covariate representing the impact of unknown sources of error based on the simulated data. Another option involves use of regularized regression techniques like LASSO [21], which impose a tuning parameter on model coefficients in order to shrink weak or null β estimates to 0. This “shrinkage” step effectively removes predictors that are statistically significant but only weakly associated with the outcome. Such approaches have several advantages for both inference and prediction, especially when paired with cross-validation and resampling procedures (e.g., bootstrap LASSO [35]), including a reduction in false-positive findings and protecting against model-overfitting.

Alternative strategies, such as Bayesian analyses providing a priori estimates that the association is falsely positive [36], are occasionally discussed in literature but rarely implemented, likely due to their complexity and strong assumptions. An emerging body of work suggests that augmenting *p*-value information with functional annotation when selecting hits for validation may improve reproducibility. These characteristics, summarized by Gorlov et al. [37] into a cogent “reproducibility index” used in genomics, range from practical (e.g., allele frequency) to biological (e.g., whether the protein encoded by the gene containing the variant of interest is a receptor or a transcription factor). Such criteria may be easily adopted for metabolomics research. However, a major challenge for metabolomics investigations lies in prioritizing features with known mechanisms of action at the expense of truly novel discoveries. More generally, the utility of incorporating functional data depends on the strength of prior evidence [38], and is expected to increase with future molecular and biochemical advances.

#### 2.1.3. Validation beyond Single Features

The increasing affordability of next-generation technologies has prompted re-evaluation of validation practices beyond single feature associations. That is to say, by prioritizing whole genes in genomics, differentially methylated regions of the genome in epigenomics, clusters of RNA transcripts in transcriptomics, groups of metabolites in metabolomics, and more recently, networks for multiple ‘omics variables. Validation of such integrative analyses requires internal cross-validation approaches to ensure stability and consistency in the ‘omics clusters. Many of these approaches follow those implemented in the field of genomics, but have been successfully applied to metabolomics or multiple-‘omics datasets. These include stability analyses to evaluate the optimal number of clusters [39], sample bootstrapping to ensure the stability of clusters [40,41], as well as qualitative assessment of whether key clusters are preserved across the validation attempts prior to external validation. We note that these analyses often involve exploratory unsupervised learning algorithms, for which there are no traditional power calculations. As this area expands, additional research is required to fill this chasm.

### 2.2. Selecting a Population for Validation

Once findings from the discovery stage have been prioritized for validation, considerations of the appropriate population for follow-up tests become paramount. The ideal choice of population lies between the Scylla of split-sample analysis (i.e., a priori splitting the original cohort into discovery and validation subsets) and the Charybdis of choosing a follow-up cohort that is so systematically different from the discovery one that any chances for successful validation are diminished.

In the first scenario, dividing the original sample into two subsamples and requiring statistical significance at a given level α in each of the subsets yields lower statistical power than analyzing the entire dataset at the α^2^ level [31]. This approach is usually undertaken when choosing a truly independent external cohort is not feasible, which occurs in studies of rare phenotypes or unique populations. In such cases, iterative resampling approaches, such as one by Kang et al., successfully implemented in GWAS studies [42]; or K-fold cross-validation, which has proven utility for metabolomics [43] may be employed to optimize statistical power. However, the performance of such methods is contingent on the proper choice of parameters, such as the ratio of the discovery to validation subsets.

In the second scenario of selecting an independent validation population, all efforts must be undertaken to harmonize the phenotype definitions and other experimental conditions across the discovery and validation samples. To reduce systematic variation, statistical models used in the validation stage should also mirror those used in the discovery stage as closely as possible. Unaccounted sources of variation between samples, whether due to discrepant trait measurements, covariate adjustment, or population characteristics, can lead to falsely negative results in the validation stage. Differences between discovery and validation populations need to be documented and, if possible, accounted for statistically. Importantly, criteria for reconciling phenotypic definitions must be established prior to analysis to avoid cherry-picking only the definitions that yield statistical significance in the validation stage.

Special caution has been applied to validation efforts across different ethnic groups. This is intuitive and imperative for genomics studies. For metabolomics, the importance of similarity of race/ethnic distributions in discovery and validation populations has been debated. On the one hand, differences in genetic architecture undoubtedly affect multiple aspects of physiology and disease etiology, and accounting for such differences will identify metabolomics profiles associated with a biological outcome that are independent of race/ethnicity. In this scenario, researchers may be interested in identifying a biomarker that performs well in a specific racial or ethnic group, such as the example of ethnicity-specific serum biomarkers of coronary artery disease [44], or the investigator may be interested in understanding specific mechanisms underlying a certain biological condition, which may differ by racial or ethnic background whether it be due to underlying psychosocial or genetic pathways. On the other hand, many metabolomics analyses seek to identify population-level biomarkers of disease risk or prognosis used in clinical settings. In these instances, the utility of a valid biomarker should be apparent across racial and ethnic subgroups and thus, a validation population need not be similar to the discovery population in terms of race/ethnicity. The decision of matching on or accounting for race/ethnic distribution for validation studies hinges on the aims of the original study, which should be clearly articulated up front.

### 2.3. Interpreting Findings

What constitutes a validated finding? As the data landscape expands and statistical methods evolve, this question is critical to the success of ‘omics science. Generally, a validated finding refers to an association between some feature and a phenotypic trait, where sufficient evidence exists to support a causal effect of the former on the latter. Sufficient causal evidence, in turn, most commonly refers to reproducibility as defined using established guidelines [45]. We propose additionally considering evidence within the context of Hill’s causal inference principles [5]. For instance, it is becoming commonplace to include data from bioinformatics databases to support biological plausibility, or use animal models and/or cell cultures to generate experimental evidence. The latter may be especially helpful in situations where establishing reproducibility proves challenging due to a lack of suitable validation cohorts, such as in some studies of exposure to unique pharmaceutical agents. (Nota bene, validation afforded by these functionally-driven approaches have been summarized by Gallagher et al. [46] and have potential to augment the lessons learned from statistical tests of the same feature-trait association across discovery and validation cohorts.)

In addition to the biological plausibility and experimental evidence guidelines discussed above, several other Hill’s postulates can be used to contemplate the quality of causal evidence supporting a given association. Consistency is directly related to reproducibility: are the same findings repeatedly observed in different places, people, circumstances and times? Strength of the association refers to its magnitude, and distinguishes between mere statistical significance (easy to obtain given large enough sample sizes) and a meaningful effect, the latter of which will require longer-term follow-up studies in the field of metabolomics to understand how differences in a given metabolite concentration related to future health outcomes. Coherence situates the new finding within the existing body of evidence, and finally, considering the criterion of specificity can elucidate the etiologic architecture of a trait. While Hill himself warned against using his criteria as a checklist for causality, they nonetheless provide a useful framework for evaluating evidence generated by discovery and validation studies.

With the growing relevance of direct-to-consumer profiling [47] in the wake of precision nutrition/precision medicine [48], another meaning of validation has come into sharper focus: how well does a feature predict trait development and/or how well does it discriminate between affected and unaffected individuals? While the numerous ethical considerations pertaining to such efforts are beyond the scope of this review, it is important to note that predictive validity extends beyond mere reproducibility or functional validation, and should be assessed using appropriate metrics that would be relevant to clinical prediction (area under the receiver operating characteristic curve, prediction *r^2^*, and others). Currently, the predictive power of ‘omics findings lags far behind reproducibility, likely due to insufficient sample sizes to detect all true effects as well as other relevant factors that explain trait variance (e.g., biological interaction). It is important to remember that successful validation – as defined by establishing reproducibility and, if possible, functional relevance – represents a necessary but not sufficient step towards using ‘omics findings to predict disease risk and, ultimately, serve as a way to identify viable avenues to improve health outcomes.

### 2.4. Metadata

In a survey conducted by *Nature* in 2016, 90% of scientists responded that there was either a “slight” or “significant” crisis of reproducibility in science [49]. This narrative has risen to prominence in recent years [50], emphasizing the importance of independently verifiable scientific discoveries. Metabolomics findings are fraught with a unique set of challenges given technical differences in analytical instrumentation (nuclear magnetic resonance spectroscopy vs. mass spectrometry, the latter of which is paired with either gas or liquid chromatography or both) and biospecimen matrix (even among studies using blood, harmonizing data from serum vs. plasma is challenging due to the type of anticoagulant used for plasma samples), as well as differences that stem from the type of platform (untargeted vs. targeted) driven by the research question. These issues, in conjunction with the high level of variability in the metabolome, make it challenging to compare and contrast metabolomics data across studies. Thus, the bare minimum scientists can do, it seems, is to be completely transparent by publishing metadata, or data about the data (“annotation”), alongside scientific findings.

In 2007, the Metabolomics Standards Initiative (MSI) put forth a set of recommendations on minimal reporting standards for the laboratory chemical analysis, as well as data analysis [51]. These standards are used as guidelines for annotation of metabolomics metadata, and continue to evolve. At present, key annotations comprise sample preparation, experimental analysis, quality control, compound identification, and data pre-processing, and focus on mass spectrometry and chromatography-based nuclear magnetic resonance spectroscopy due to the popularity of these techniques in metabolomics [52]. As additional instrumental (e.g., use of capillary electrophoresis based mass spectrometry), bioinformatics, and statistical techniques emerge, proper annotation of the scientific data to ensure metadata quality will become ever more important. As discussed by Ferreira et al. [53], quality control of metadata is no small task and will likely require automated mechanisms [54] to assess the quality of metadata. Metadata may include annotations of specific biological pathways that are relevant to the study findings, and/or a description of processes by which the data were collected (e.g., how the data was collected, by whom and when, storage procedures, biospecimen processing techniques, and software used) and processed (e.g., bioinformatics pipelines and decisions made regarding artefacts of the data). Ultimately, such semantics allow for the harmonization of data from different sources, facilitate reproducible research, and allow for more confident interpretation and dissemination of findings.

## 3. Summary of Considerations for Metabolomics

For metabolomics specifically, replication and validation is critical and challenging given the higher degree of variability in metabolites, and the vulnerability of metabolite concentrations to lifestyle and environmental factors that are of markedly less concern in other ‘omics fields, notably genomics. Regardless, several topics discussed in this review are relevant to metabolomics analyses, including the need for confirmation via replication in independent study samples, the importance of internal and external validation (the latter of which requires selection of an appropriate study population to reduce systematic variability), selection of appropriate methods to correct for multiple comparisons, and use of criteria beyond *P*-values to identify “hits” – namely, consideration of the magnitude of effects. Additionally, while not formally discussed herein, researchers should be aware of statistical phenomena that may impact the magnitude of effects detected in initial discovery studies (e.g., “Winner’s curse”), and employ *a priori* considerations of statistical power.

## 4. Future Directions: Integration of Multiple ‘Omics

The use of multiple ‘omics datasets is becoming increasingly popular in all facets of life science. In the biomedical sciences, genomics-metabolomics analyses using undirected network-based approaches have demonstrated capacity to recreate true biochemical pathways [55] and home in on those involved in disease etiology [56]. While such studies require a more complex set of strategies to deal with replication and validation, probing ‘omics profiles from both ends of the central dogma of molecular biology—genetics on the one side, and metabolomics on the other side—to provide valuable information on modifiability of the mechanisms represented by the networks. That is to say, in a network where a SNP explains substantial variation in the cluster of molecular variables, we can infer that pathway of interest stems from genetic variation, and that metabolites are mediators of the process. Given the non-modifiability of genetics, identifying determinants of networks comprised predominantly of SNPs is less relevant to preventive action but may be informative for surveillance. On the other hand, in a network without any genetic associations, metabolic perturbations are likely to originate from external factors and thus, may represent modifiable mechanisms underlying disease progression. Accordingly, linking environmental and/or lifestyle characteristics to these “modifiable networks” will point toward avenues for prevention. As an example, Krumsiek et al. [57] analyzed untargeted metabolomics data from 1756 men and women in the KORA study to identify metabolites that exhibit marked differences in serum concentrations. These compounds were on steroid metabolism, fatty acids (a large fraction of which were related to amino acid metabolism), oxidative phosphorylation, and purine metabolism pathways. A network-based clustering of metabolites of interest with a gender-stratified GWAS revealed gender-specific differences in metabolic networks that were predominantly explained by variability in the metabolites, as opposed to the SNPs, suggesting that gender differences in these metabolic pathways are not caused by genetic dimorphisms, but, rather, polymorphisms of the metabolome that may be influenced by the environment and/or behaviors. Future studies identifying lifestyle and behavioral correlates of such metabolic networks will shed light on specific preventive interventions that have potential to mitigate risk of chronic diseases known to be influenced by the above-mentioned biochemical pathways (e.g., type 2 diabetes, cardiovascular disease).

The above concepts are the embodiment of innovative studies that capitalize on multiple ‘omics datasets to unveil findings with meaningful implications for actionable next steps. The rapid growth of ‘omics technologies and statistical packages to analyze such data necessitates the need to establish best practices for bioinformatics and analytical workflows within each individual ‘omics field, as well as for multiple ‘omics. Given that reproducible research is the foundation upon which scientific inference is made, statistical validation and replication serve as pillars that flank the journey from hypothesis testing to informing next steps for preventive or therapeutic action.

## Figures and Tables

**Table 1 metabolites-10-00286-t001:** Best practices for replication and validation of ‘omics findings.

**Both**
● A priori calculations of the sample size required to detect a realistic effect.
● Use of publicly available ‘omics datasets and/or pursuit of collaborations with other cohorts/consortia to maximize statistical power.
● Stringent and appropriate corrections for multiple testing.
● Transparent reporting of all relevant methods (from the laboratory work to bioinformatics pipelines, to data cleaning, to formal data analysis), features, and results (including those that failed to establish reproducibility).
**Validation**
● Harmonizing data across the discovery and validation stages to reduce the likelihood of non-reproducible findings due to systemic differences.
● Inclusion of diverse datasets in validation efforts and the use of appropriate statistical methods to account for the resulting heterogeneity.
● Judicious incorporation of functional annotations and effect sizes (in addition to statistical significance) when selecting features for validation and interpreting findings.
● Distinguishing between reproducibility, functional relevance, and predictive validity, and using the appropriate metrics for each.
**Replication**
● Original (discovery) and confirmatory (replication) populations should be similar in terms of sex, age, and race/ethnic distributions.
● Use of identical laboratory procedures, data processing pipelines, and analytical approaches.

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
