# Peer review of "Find the Needle in the Haystack, Then Find It Again: Replication and Validation in the ‘Omics Era"

_metabolites, 2020, doi:10.3390/metabo10070286_

Round 1
Reviewer 1 Report
1. This is a fairly good but a very generic review of the replication and validation problem in the Omics field. The authors have spent a good amount of space on reviewing the issues in the Genomics area, which has already been done numerous times in Genomics/Transcriptomics journals. This is reflected from the reference list as well where the metabolomics specific studies are very low in numbers as compared to other Omics studies.
2. The last section where provide suggestions for improving the replicability of Metabolomics by integrating other OMICS should is pretty generic as well and lacks novelty. They should expand this section and give some specific examples/suggestions for the Metabolomics researchers.
3. The metadata is even more important in Metabolomics and they should give suggestions (or references) on how one should capture this well in their studies for good replicability later.
Author Response
- Comment: This is a fairly good but a very generic review of the replication and validation problem in the Omics field. The authors have spent a good amount of space on reviewing the issues in the Genomics area, which has already been done numerous times in Genomics/Transcriptomics journals. This is reflected from the reference list as well where the metabolomics specific studies are very low in numbers as compared to other Omics studies.
Response: We are grateful to the reviewer for their time vetting this work, and for their suggestions. In light of this comment, we have re-written large portions of this review to discuss major concepts as they relate to metabolomics. We believe this is reflected by the new literature cited as well. Major changes are highlighted in yellow. - Comment: The last section where provide suggestions for improving the replicability of Metabolomics by integrating other OMICS should is pretty generic as well and lacks novelty. They should expand this section and give some specific examples/suggestions for the Metabolomics researchers.
Response: In our revised manuscript, we now discuss replicability and validity specifically for metabolomics analyses throughout the body of the text, with some historical and contemporary examples from genomics to provide context for why certain approaches are used in metabolomics followed by discussion on whether such approaches are appropriate. In the last section of this review, the goal is not to propose integrative ‘omics studies as a way to improve metabolomics replicability. Rather, the aim of this last section is to bring to light the most recent efforts in this arena, and to discuss implications and future directions of this work. In response, we now provide a specific example of how integrative multi-‘omics analyses may be used to inform next steps for preventive or therapeutic action. To make this section more specific, we have added the following text:
Page 20-21, lines 420-440: As an example, Krumsiek et al. (57) analyzed untargeted metabolomics data from 1756 men and women in the KORA study to identify metabolites that exhibit marked differences in serum concentrations. These compounds were on steroid metabolism, fatty acids (a large fraction of which were related to amino acid metabolism), oxidative phosphorylation, and purine metabolism pathways. A network-based clustering of metabolites of interest with a gender-stratified GWAS revealed gender-specific differences in metabolic networks that were predominantly explained by variability in the metabolites, as opposed to the SNPs, suggesting that gender differences in these metabolic pathways are not caused by genetic dimorphisms, but rather, polymorphisms of the metabolome that may be influenced by the environment and/or behaviors. Future studies identifying lifestyle and behavioral correlates of such metabolic networks will shed light on specific preventive interventions that have potential to mitigate risk of chronic diseases known to be influenced by the above-mentioned biochemical pathways (e.g., type 2 diabetes, cardiovascular disease).
The above concepts are the embodiment of innovative studies that capitalize on multiple ‘omics datasets to unveil findings with meaningful implications for actionable next steps. The rapid growth of ‘omics technologies and statistical packages to analyze such data necessitates the need to establish best practices for bioinformatical and analytical workflows within each individual ‘omics field, as well as for multiple ‘omics. Given that reproducible research is the foundation upon which scientific inference is made, statistical validation and replication serve as pillars that flank the journey from hypothesis testing to informing next steps for preventive or therapeutic action.
- Comment: The metadata is even more important in Metabolomics and they should give suggestions (or references) on how one should capture this well in their studies for good replicability later.
Response: This point is well-taken. We now include a section on Metadata (Section 2.3, pages 18-19, lines 361-391)
Reviewer 2 Report
Extremely well written review, only a few typos (highlighted on the attached) to potentially change.

Author Response
- Comment: Extremely well written review, only a few typos (highlighted on the attached) to potentially change.
Response: We are grateful for the reviewer’s critical eye for detail. We have corrected the typos.
Round 2
Reviewer 1 Report
Thank you for incorporating my suggestions into your manuscript.